# Directed Network Disassembly Method Based on Non-Backtracking Matrix

**Jinlong Ma** [1,2], **Peng Wang** [1,2] **and Huijia Li** [3,*]

1   School of Information Science and Engineering, Hebei University of Science and Technology, Shijiazhuang 050018, China
2   Hebei Technology Innovation Center of Intelligent Internet of Things, Shijiazhuang 050018, China
3   School of Science, Beijing University of Posts and Telecommunication, Beijing 100876, China
*   Correspondence: lihuijia0808@gmail.com

**Abstract:** Network disassembly refers to the removal of the minimum set of nodes to split the network into disconnected sub-part to achieve effective control of the network. However, most of the existing work only focuses on the disassembly of undirected networks, and there are few studies on directed networks, because when the edges in the network are directed, the application of the existing methods will lead to a higher cost of disassembly. Aiming at fixing the problem, an effective edge module disassembly method based on a non-backtracking matrix is proposed. This method combines the edge module spectrum partition and directed network disassembly problem to find the minimum set of key points connecting different edge modules for removal. This method is applied to large-scale artificial and real networks to verify its effectiveness. Multiple experimental results show that the proposed method has great advantages in disassembly accuracy and computational efficiency.

**Keywords:** directed network dismantling; non-backtracking matrix; spectral partition; minimal dismantling set





## 1. Introduction

In complexity science, a network (denoted by $G = (V, E)$ in graph theory) is composed of a node set $V$ consisting of $n$ nodes and an edge set $E$ consisting of $m$ edges between the nodes. Many real-world networks such as the Internet, WWW, large-scale power networks, transportation networks and interpersonal networks can be modeled in this concise way [1]. Using this method, these networks can be regarded as a collection of nodes with independent characteristics interconnected with other individuals. Each individual is regarded as a node in the network, and the connection between nodes is regarded as the edge of the network. This abstract method can intuitively show the topology of the real network, and also provides an effective research method for understanding the state and the function of the real network [2].

However, with the continuous development of technology and society, epidemic viruses [3], computer viruses [4], misinformation [5], or corruption [6] have more serious negative effects in the human world. However, removing or deactivating a part of the key nodes through the network dismantling method in the network to decompose the network into several isolated sub-parts can effectively protect the robustness of the network, control the dynamic behavior of the network, and curb the negative effects in the network mentioned above. Previous studies proved that this method to remove or deactivate the key nodes can effectively curb the spread of epidemics in the population [7], prevent the spread of misinformation through social networks [8] and prevent the spread of viruses in computer networks [9]. Some studies on complex networks choose a set of node subsets $S$ in the network with an optimal method, and explore the influence of removing $S$ on the network characteristics. For example, exploring how the maximum connected (strong) subset of the network will change after removing $S$, in the example of epidemics or network

viruses transmission, if $S$ is isolated or infected first, the impact on the speed of virus transmission can be determined [10]?

In the actual situation, it will produce a certain cost consumption $C$ when selecting and removing the node subset $S$ in the network. In the epidemic propagation model, the vaccination of the node requires a certain socioeconomic cost. Removing different node sets in the computer virus or public opinion propagation network consumes different resources in the actual situation. Therefore, a combinatorial optimization problem is generated, whereby under the influence of the constraint removal node subset $S$ in the network, the removal cost $C$ is minimized. Additionally, removing nodes will destroy the network structure and affect the function of the network, so it is necessary to remove as few nodes as possible and find a set of nodes with the lowest removal cost $C$, that is, the minimum disassembly set (we consider the most common situations where the disassembly cost is the unit cost; the minimum disassembly set is the set with the least number of nodes) and remove it. After the network is decomposed into multiple sub-parts that are not connected, network disassembly is achieved [11]. Finding the minimum disassembly set is an NP-hard problem [12]. For this kind of problem, only effective approximation algorithms or heuristic algorithms can be found at present. For the disassembly problem, there have been some recursive algorithms based on the degree or centrality of the nodes. For example, in 2015, a heuristic algorithm called 'collective' influence (CI) [13] was proposed, which determines the ownership of the nodes in an undirected random network according to the degree of nodes and the degree of local neighbor nodes; in 2016, Alfredo Braunstein proposed a three-segment minimum sum (hereafter referred to as the Min-Sum) method for dismantling large random undirected networks [11]; for large undirected random networks, this method is a more effective dismantling algorithm. In 2019, Ren proposed a general network disassembly (hereafter referred to as the GND) method for undirected weight networks [14]. In addition, some relatively new disassembly methods and analyses of disassembly [15,16] are provided, such as the disassembly algorithm based on the message passing model (2020) [17] and the sensitive disassembly method of neighbor connection (2020) [18].

Most of the existing disassembly algorithms are carried out in undirected networks, while there are few disassembly algorithms for directed networks. However, in many real-world networks (such as WWW networks, acquaintance relationships, network email networks, text association networks and article citation networks, etc.), the edges between nodes are unidirectionally connected, and there is no mutual relationship in undirected networks [19]. The existing disassembly methods are sometimes not suitable for the disassembly of these directed networks. Compared with disassembly in the undirected network, disassembly in the directed network needs to consider the direction of the edges between the nodes in the network. For example, in a public opinion network, when an incoming node (Innode) connected by a directed edge $e$ (also known as an arc in graph theory) is a communicator, the outgoing node of this edge (Outnode) is not likely to be propagated by the node. When applying the undirected network-based disassembly method to disassemble the network, this one-way relationship between nodes is sometimes ignored, resulting in the removal of this edge $e$, and causing unnecessary disassembly to affect the disassembly effect.

The influence of the internal mechanism of this network on the disassembly is ignored in the traditional disassembly method. To solve this problem, the non-backtracking matrix representing the edge adjacency relationship is applied as the operator of spectral division, whereby it retains the directionality of the node relationship, and combines the disassembly of the directed network with the spectral division method. An improved spectral disassembly method for directed networks (hereafter referred to as the DIR method) is proposed; the edges in the directed network are directly applied as the disassembly unit, and the strongest connected subgraph of the directed network is used as the disassembly subject. The spectral characteristics of the edge adjacency matrix (non-backtracking matrix) are applied to the bipartition of the edge modules in the maximum strongly connected subset.

The overlapping node set of the node sets connecting the two edge modules is then found as the minimum disassembly set to disassemble the network until the disassembly scale reaches the specified disassembly scale of the network (the maximum number of nodes in the strongly connected subgraph). The excellent characteristics of the non-backtracking matrix can be made full use of by using the DIR method, and the DIR method can greatly protect the topology of the network during the disassembly process. Furthermore, it is verified by experiments that the DIR method is suitable for the disassembly of directed networks. Finally, the influence of different disassembly methods on the network structure is analyzed by analyzing the changes of network indexes such as the clustering coefficient, assortativity coefficient and modularity function in the disassembly process, and it is verified that the application of edge module partition to disassemble the network can greatly retain the structural information in the networks.

## 2. Related Works

As described in the previous section, many network dismantling methods have been proposed in recent years. Next, I will introduce two methods that are compared with this article, namely the GND algorithm and Min-Sum algorithm.

GND method. This method considers the case where the node removal cost is equal to the node weight and is not a unit cost. First, perform spectral division of the Laplacian matrix for which the operators are node weights of the network. After the division is successful, the node weight coverage algorithm is applied to the edges connecting different divisions to find the minimum weight point set that can divide the network, so as to find the minimum cost disassembly set. Compared with the previous algorithm, GND is more general and applicable. It considers the influence of node weights in undirected networks on the network disassembly problem. However, the operator in the GND method is a node adjacency matrix.

Min-Sum method. Braunstein et al. proposed a three-stage Min-Sum algorithm to dismantling networks. They first decycle a network with a variant of the Min-Sum message passing algorithm. After all cycles are broken, they break the remaining trees into small components until the largest component is smaller than the desired threshold. Finally, they refine the node set of network dismantling by moving some of them back to the original network. However, this kind of method tends to delete irrelevant nodes during the loop removal step and then moves them back to the original network in the following node re-inserting step, which reduces the disassembly efficiency.

However, when analyzing directed networks, most spectral methods using node adjacency matrices will use symmetric adjacency matrices to make the network undirected [20]. This processing method will inevitably lose some information in the network [21], resulting in the search process for the minimum set of nodes to be disassembled in the directed network which will add some unnecessary nodes and cause unnecessary disassembly.

## 3. Preliminaries

In this section, we will provide a simple disassembly flow chart and introduce the knowledge of non-backtracking matrices so that readers can understand the proposed method more easily.

### 3.1. Model

As shown in Figure 1, by applying the DIR method to disassemble the directed network in the figure, according to the spectral characteristics of the non-backtracking matrix, the edge of the directed network is divided into two different red and blue modules; overlapping nodes 5 and 6 that connect these two different edge modules were found. By removing nodes 5 and 6 and disconnecting the edges connected to them, the directed network can be divided into two disconnected sub-parts. Compared with the previous disassembly method, this disassembly method for removing overlapping nodes of edge modules requires fewer disassembly steps, does not need to find the minimum node cover-

age set, and the corresponding disassembly cost is lower (the nodes found by the traditional decomposition method are 5, 6, 7, 12), which is more suitable for directed networks.

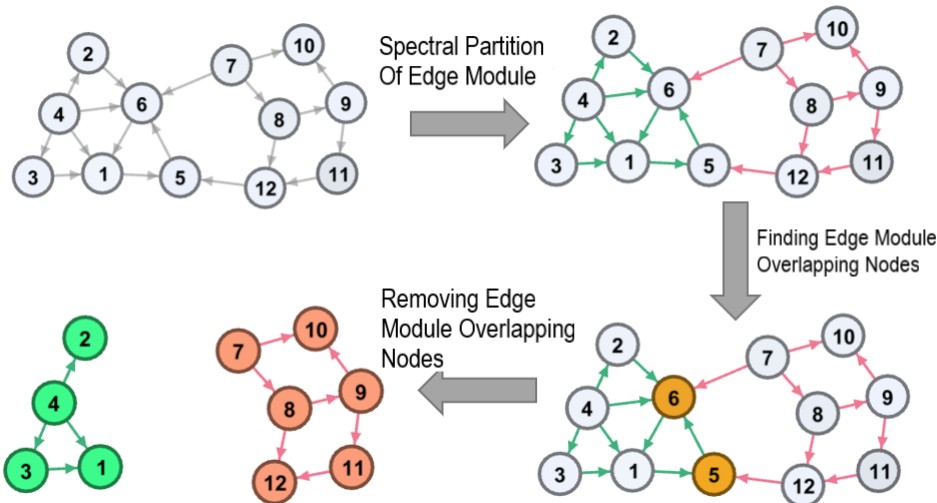

**Figure 1.** Directed network disassembly flow chart.

### 3.2. Non-Backtracking Matrix

In a directed network $G$, $i$, $j$, $k$ and $l$ are all nodes in $V$, according to the definition of non-backtracking random walk, but only when $j = k$ and $i \neq l$, directed edge $i \rightarrow j$ is connected to another directed edge $k \rightarrow l$. In a directed network, $B$ is a $m * m$ non-backtracking matrix. This non-backtracking matrix is used to represent the adjacency relationship of edges in a directed network, defined as

$$B_{i \rightarrow j, k \rightarrow i} = \begin{cases} 1, & \text{if } j = k \text{ and } i \neq l \\ 0, & \text{other cases} \end{cases} \tag{1}$$

The non-backtracking matrix $B$ is different from the adjacency matrix $A$, where $B$ takes each directed edge as an element, and represents the adjacency relationship between the edges in the matrix; therefore, it is also called the edge adjacency matrix. The excellent properties of the non-backtracking operator have been shown above [22], and the spectral characteristics of the non-backtracking matrix have better performance in the network than the node adjacency matrix $A$ or other matrices, especially in terms of the strong separation of its second eigenvector for the network structure division. At the same time, directed networks in the real world tend to have relatively sparse structures and large scales. The non-backtracking matrix $B$ also performs well in sparse networks compared to the node adjacency matrix $A$. The adjacency matrix of the edge, $B$, stores the relationship between the edges in the network, and is not sensitive to the information of the nodes in the network so as to avoid the tendency to remove the nodes with a relatively large degree during dismantling and cause damage to the connected subset of the network [23], thus retaining the structural information in the directed network to the greatest extent. It is also proved by experiments that applying the non-backtracking matrix to disassemble the one-way connection relationship of the edges in the directed network can reduce the disturbance of the node's topology information to the selected disassembly node set, and effectively avoid the problem of network information loss when directly using the directed network adjacency matrix as the spectral algorithm operator.

### 4. Method

In this section, we propose a method that combines edge module partition with network disassembly to construct a network disassembly algorithm in the directed networks. The non-backtracking matrix is used to store the adjacent information of the edges, and

the non-backtracking matrix is used as the operator to construct the minimum number of edges in the disassembly function. The edge module is divided by solving the approximate second eigenvector of the function matrix; after the division, the minimum number of edge sets connecting the different edge modules and the node set where the modules overlap in the edge sets are determined. By removing this node set, the connection between different modules is destroyed, and disassembly is finally achieved.

### 4.1. Disassemble the Objective Function

In this section, we consider the general case of dividing a network in two modules of equal size according to the nature of the edges, minimizing the number of edges between two different modules. The non-backtracking matrix is used to store the edge adjacency information in the directed network, because in the disassembly problem, we will eventually remove all overlapping nodes on different edge modules, and the weight of the edge does not affect the selection of the minimum node set; therefore, we set the weight of each edge as the unit weight. We divide $m$ edges in the edge network into two equal-sized $\frac{m}{2}$ modules according to the corresponding characteristic. We define an index variable $s_{i \to j} \in R^m$ for any directed edge $i \to j, i, j \in N$ in the network, and assume that if this edge $i \to j$ belongs to partition module 1, then $s_{i \to j} = 1$; if edge $i \to j$ belongs to partition module 2, then $s_{i \to j} = -1$. So, we obtain

$$\frac{1}{2}\left(s_{i \to j}s_{j \to k} + 1\right) = \begin{cases} 1, & \text{If two connected edges} \\ & i \to j, j \to k(i \neq k) \text{ belong to} \\ & \text{the same edge module} \\ 0, & \text{other cases} \end{cases} \tag{2}$$

Equation (2) is used to determine whether two connected edges belong to the same module. Combined with Equation (2), we use the non-backtracking matrix as an operator to obtain the objective function of the minimum number of disassembled edges, which is used to find a set of edges that connect two different modules with the smallest number:

$$\begin{aligned} \min : R &= \sum_{i \to j, k \to l} B_{i \to j, k \to l} - \sum_{i \to j, k \to l} \frac{1}{2}\left(s_{i \to j}s_{k \to l} + 1\right)B_{i \to j, k \to l} \\ &= \frac{1}{2}\sum_{i \to j, k \to l}\left(1 - s_{i \to j}s_{k \to l}\right)B_{i \to j, k \to l} \\ &= \frac{1}{2}\sum_{i \to j, k \to l}\left(d_{i \to j}\delta_{k \to l, i \to j} - B_{i \to j, k \to l}\right)s_{i \to j}s_{k \to l} \\ &= \frac{1}{2}s^T B's \end{aligned} \tag{3}$$

$$\text{s.t.} \begin{cases} 1^T s = 0, \\ s_{i \to j} \in R, i \to j \in E \end{cases} \tag{4}$$

where $B' = D_B - B$, $D_B$ is a diagonal matrix, $(D_B)_{i \to j, i \to j} = \sum_{k \to l} B_{i \to j, k \to l}$. Equation (3) represents the difference obtained by the logarithm of the minimized total connected edges minus the logarithm of the edges connected inside the edge module. When two connected edges are divided into different edge modules, $s_{i \to j}s_{j \to k} = -1, B_{i \to j, j \to k} = 1$, the nodes connecting the two edges needs to be removed; on the contrary, when two connected edges are divided into the same edge module, the nodes connecting the two edges do not need to be removed. Finally, the set of nodes that need to be removed corresponding to the set of partitions that minimize $R$ is the minimum disassembly set. We specify the number of nodes whose disassembly cost is the minimum disassembly set.

$1^T s = 0$ in Equation (4) ensures that the two modules are of equal size. Unfortunately, this optimization problem is an NP-hard problem. For this problem, the approximate solution can be found by relaxing constraint $s_{i \to j} \in \{-1, 1\}$ to $s_{i \to j} \in R$. According to the Courant--Fisher theory[24], the solution of this relaxation constraint minimum

optimization problem can be found by analyzing the eigenvector $v^{(2)}$ corresponding to the second smallest eigenvalue $\lambda_2$ of $\boldsymbol{B}'$. So, if node $j$ connects two edges $i \rightarrow j, j \rightarrow k(i \neq k)$ corresponding to the value of the second smallest eigenvector, one of the second smallest eigenvectors are non-negative $\left(v^{(2)}_{i \rightarrow j} \geq 0\right)$, and the other's second smallest eigenvector is negative $\left(v^{(2)}_{i \rightarrow j} < 0\right)$; this node will be removed. Removing all such nodes in the network can decompose the network into two sub-parts.

### 4.2. Divide Vector

Because the large-scale network has many edges, its corresponding second eigenvector of $\boldsymbol{B}$ is difficult to obtain accurately [25]. The traditional power-law iterative model is applied to perform a simple and refined approximation algorithm for the second smallest eigenvalue. Matrix $\boldsymbol{B}'$ has $m$ real non-negative eigenvalues $\lambda_1 \leq \lambda_2 \leq \ldots \leq \lambda_m$, and the corresponding eigenvectors are $v^{(1)}, v^{(2)}, \ldots, v^{(m)}$, which are orthonormal bases in $\boldsymbol{R}^m$ space. We define the maximum degree of elements in matrix $\boldsymbol{B}$ as $d_{\max}$, $x, y$ represents the row and column of the matrix, and the upper bound of the spectrum can be obtained by calculating the 1-norm.

$$
\begin{aligned}
\lambda_m &\leq \max_{\|v\|_1=1} \|(\boldsymbol{D_B} - \boldsymbol{B})v\|_1 \\
&= \max_{\|v\|_1=1} \sum_{x=1}^{m} \left| v_x \sum_{y=1}^{m} \boldsymbol{B}_{xy} - \sum_{y=1}^{m} v_y \boldsymbol{B}_{xy} \right| \\
&\leq \max_{\|v\|_1=1} \sum_{x=1}^{m} \sum_{y=1}^{m} |v_x \boldsymbol{B}_{xy}| + \sum_{x=1}^{m} \sum_{y=1}^{m} |v_x \boldsymbol{B}_{xy}| \\
&= \max_{\|v\|_1=1} \|\boldsymbol{B}v\|_1 + \|\boldsymbol{B}v\|_1 \\
&= 2d_{\max}
\end{aligned}
\tag{5}
$$

The upper bound of the spectrum calculated by Equation (5) is $\lambda_m \leq 2d_{\max}$. In order to calculate the approximate second eigenvector, we calculate the matrix $\boldsymbol{H} = 2d_{\max} - \boldsymbol{B}'$, which has the same eigenvector as $\boldsymbol{B}'$. Therefore, the corresponding eigenvalue is now converted into the calculation $0 \leq \xi_m = 2d_{\max} - \lambda_m \leq \ldots \leq \xi_1 = 2d_{\max}$, in which the eigenvector $v^{(2)}$ corresponding to the second largest eigenvalue $\xi_2$ is calculated. Then, we find the eigenvector of $\boldsymbol{H}$ corresponding to the eigenvalue $\lambda_2$ using the following power-law iterative algorithm.

Algorithm 1 can find an approximate eigenvector corresponding to $\lambda_2$; we can use our orthogonal eigenvector basis to represent any random vector $v = \sum_{i=1}^{m} \varphi_i v^{(i)}$; the second step of the algorithm can guarantee $\varphi_1 = 0$ and $\varphi_2 \neq 0$. Finally, by multiplying the vector $v$ by the linear operator $\boldsymbol{H}^k$, we obtain

$$
\boldsymbol{H}^k v = \sum_{i=2}^{m} \varphi_i v^{(i)} \propto \varphi_2 v^{(2)} + \sum_{i=3}^{m} \varphi_i \left( \frac{\xi_i}{\xi_2} \right)^k v^{(i)}
\tag{6}
$$

Since $\lambda_3 > \lambda_2$, there is $\left| \frac{\xi_i}{\xi_2} \right| < 1$, and we obtain $\varphi_i \left( \frac{\xi_i}{\xi_2} \right)^k v^{(i)} \rightarrow 0$. When the scale of the index $k$ (the number of iterations) of the operator $\boldsymbol{H}$ is $O\left(\log(m)^{1+\varepsilon}\right)$, $v$ tends to be the expected value $E\left[\left| \lambda_2 - \frac{v^T \boldsymbol{B} v}{v^T v} \right|\right] \rightarrow 0$ of the eigenvalue $\lambda_2$ corresponding to $\boldsymbol{B}'$, where $m$ is the number of edges of the real network.

---

**Algorithm 1:** Approximate feature vector algorithm

---

input:     Non-backtracking matrix $\boldsymbol{B}$, network edge number $m$, $v_1 = (1, 1, \ldots, 1)^T$
output:   Approximate second eigenvector $v$
1: Randomly select vector $v$ on the unit sphere;
2: $v \leftarrow v - \frac{v^T v}{v^T v_1} \cdot v_1$;
3: For $i = 1$ to $\tau(m)$;
4: $v \leftarrow \frac{Hv}{\|Hv\|}$;
5: End for;
6: Return $\boldsymbol{v}$.

---

### 4.3. Directed Network Disassembly

Algorithm 2 provides a recursive solution that can repeatedly disassemble a network to a specified scale. The number of nodes in the maximal strongly connected subset *GSC* is defined as the disassembly scale. In this algorithm, we intend to disassemble the directed network until the disassembly scale is smaller than the target scale *C*. The above algorithm is also defined according to this idea. The input of the algorithm is the node-edge topology of the directed network. The final output is the minimum node disassembly set and the required disassembly cost when the directed network is disassembled to a specified scale; in the first step of the algorithm, the maximal strongly connected subset of the network is taken as the disassembly subject of the directed network, which can filter out the nodes and edges in the network that are not related to network disassembly; this can further improve the disassembly efficiency of the directed network. The selection of the strongly connected subset as the disassembly subject can be directly compared with the connected subset of the commonly used undirected network, which can avoid undirectional networks to meet the undirected network disassembly conditions and cause redundant disassembly. The process is controlled by judging the size of the strongest connected subset of the network; the minimum disassembly set and disassembly cost are initialized to 0 in step 2; the Laplacian matrix of the non-backtracking matrix of the maximum strongly connected subgraph is generated in step 3 for the next division of the edge module; in the fourth step, the eigenvector corresponding to the second eigenvalue is obtained by calculating $H = 2d_{\max} - \boldsymbol{B}'$ and applying the eigenvector approximation algorithm, which is used to divide the edge module; the overlapping node set between edge modules is found and removed in the entire network $\boldsymbol{G}$ in step 5 and 6; the node to be removed is added to the disassembly set and the maximum strongly connected subset and disassembly set of the network in step 7 are updated; the minimum disassembly set and disassembly cost are updated in step 8; whether the maximum strongly connected subset size of the network reaches the target disassembly size is determined in step 9. This recursive algorithm can obtain the set of nodes that disassemble the directed network into a minimum set of connections between different edge modules of a specified size.

---

**Algorithm 2:** Directed network disassembly algorithm (DIR method)

---

input:     Network $G$
output:   Minimum disassembly set $L_s$, minimum disassembly cost $c$
1: Select the maximum strongly connected subgraph *GSC* in the network and calculate its non-backtracking matrix $\boldsymbol{B}_{GSC}$ according to Equation (1);
2: Initialize $L_s$, $c$ to 0;
3: Calculating $\boldsymbol{B}'$ corresponding to $\boldsymbol{B}_{GSC}$ by Equation (3);
4: Use algorithm 1 to obtain the division vector v and divide the maximum strongly connected subgraph into two edge modules;
5: Find the edge set connecting the two edge modules and create a partition subgraph;
6: Find the overlapping node set S in the partitioned subgraph;
7: Remove $S$ from network $G$ and update network $G$;
8: Merge $S$ into $L_s$ and update $L_s$, $c$, and $\boldsymbol{B}_{GSC}$;
9: If the size of the largest strongly connected subgraph $GSC_{size}$ < target disassembly size $C$, return $L_s$ and $c$;
Otherwise, go back to step 3.

---

### 4.4. Algorithm Complexity

The time complexity of the approximate feature vector is equal to the number of iterations $\tau(m)$ multiplied by the product of matrix $\boldsymbol{H}$ and vector $v$, namely $O(\tau(m)m^2)$, where $m$ is the number of network edges.

The complexity of performing a bisection for the entire network is $O(m^2\tau(m))$. The complexity of performing another bisection on the two modules with an approximate size of $m/2$ after the division is $2 \cdot O\left(\left(\frac{m}{2}\right)^2\tau(m)\right)$. The complexity of another bisection for the four modules with an approximate size of $m/4$ after division is $4 \cdot O\left(\left(\frac{m}{4}\right)^2\tau(m)\right)$. Until $O(GSC) = 1$, the complexity of another bisection for $m/2 = 2^{\log_2(m)-1}$ modules with an approximate scale of 2 after division is $2^{\log_2(m)-1}O\left(\left(\frac{m}{2^{\log_2(m)-1}}\right)^2\tau(m)\right)$. The total time complexity is as follows:

$$
\begin{aligned}
O&\left(m^2\tau(m)\right) + 2 \cdot O\left(\left(\frac{m}{2}\right)^2\tau(m)\right) + 4 \cdot O\left(\left(\frac{m}{4}\right)^2\tau(m)\right) \\
&+ \ldots + 2^{\log_2(m)-1}O\left(\left(\frac{m}{2^{\log_2(m)-1}}\right)^2\tau(m)\right) \\
&= \sum_{i=0}^{\log_2(m)-1} 2^i O\left(\left(\frac{m}{2^i}\right)^2\tau(m)\right) = O\left(m^2\tau(m)\right)\left(\sum_{i=0}^{\log_2(m)-1} 1\right)^2 \\
&= O\left(m^2\tau(m)\log^2(m)\right)
\end{aligned}
\tag{7}
$$

The computational complexity of the dismantling recursive algorithm is $O\left(m^2\tau(m)\log^2(m)\right)$. For a sparse network, $\tau(m) = \log(m)^{1+\varepsilon}$ at moment $\varepsilon > 0$ and there is an upper bound $1/\log\left(\left|\frac{\xi_2}{\lambda_3}\right|\right)$ [14]; therefore, a better dismantling effect can be obtained. The computational complexity is $O\left(m^2\log(m)^{3+\varepsilon}\right)$ at this moment.

In the algorithm, the space required for each non-backtracking matrix is $O(m^2)$, the recursive depth is $O(\log(m))$, and the required space complexity is $O(m^2\log(m))$, where $m$ is the number of network edges.

## 5. Experimental Results

In order to verify the applicability of the DIR method in directed networks, it is used in artificial directed ER networks, BA networks and real networks, and the disassembly results are compared with two commonly used methods (GND algorithm and Min-Sum method). The dataset of Table 1 is selected in the experiment, and the experimental comparison is carried out in different artificial directed networks and large-scale real networks (for the convenience of comparison, the disassembly scale and disassembly cost in this paper are both proportional).

**Table 1.** Network dataset.

| Network Name | Number of Nodes $n$ | Number of Edges $m$ | Node Connection Probability $p$ | Average Degree |
|---|---|---|---|---|
| ER random network | 1000 | approximately equal to 10,000 | 0.01 | 10 |
| BA random network | 1000 | approximately equal to 10,000 | | 10 |
| Email-EU-core network | 1005 | 24,929 | | 24.80 |
| Weki-vote network | 8297 | 103,689 | | 12.50 |

Some scholars, e.g., Ren [14] have proved that the GND method has a higher disassembly efficiency than other algorithms such as Min-Sum and information transfer in undirected networks when the network disassembly cost is the unit cost (number of nodes) and non-unit cost (based on node degree). When the disassembly scale is the same, the

GND method has a lower disassembly cost than other algorithms, and the GND method can destroy the network structure with a smaller disassembly cost. This method has better performance than other algorithms in the disassembly of undirected networks. The DIR method is compared with the GND method and Min-Sum method, considering that the disassembly cost is the unit cost (i.e., the number of nodes).

Figures 2 and 3 are the disassembly results of different methods in different directed networks, where the corresponding curves of disassembly scale and disassembly cost are provided. The ordinate disassembly scale is the proportion of the number of nodes in the largest (strongly) connected subgraph, and the abscissa disassembly cost is the proportion of the number of nodes in the smallest disassembly set when disassembly is at the scale shown in the ordinate. As shown in Figure 2, in a dense ER random network, when the required disassembly size is less than 0.25, the disassembly cost of the DIR method is smaller than that of the GND and Min-Sum methods; in the artificial BA directed network with an average degree of 10, the disassembly cost of the DIR method is significantly lower than that of the other two methods. Additionally, in the relatively sparse real directed network (Figure 2), when the network is disassembled to the same specified scale, the DIR method has a lower cost than other methods. The reason for the difference is that methods such as GND and Min-Sum take the largest connected subgraph in the network as the disassembly subject. The DIR method takes the largest strongly connected subset of the network as the disassembly subject. When the network is dense enough, the size of the largest strongly connected subgraph and the connected subgraph in the directed network is not hugely different; however, in the relatively sparse network (such as the Weki network), the cost of applying the maximum strongly connected subgraph of the directed network for disassembly is significantly lower than that of the GND and Min-Sum methods. The experiments show that the DIR method has the advantage of lower cost in directed network disassembly, which shows the efficiency of the DIR method in directed network disassembly.

Next, we explore the impact of different disassembly methods on the network structure. By applying different disassembly methods to different networks and comparing the clustering coefficient [26] and assortativity coefficient [27] of the disassembly process network, the superiority of the DIR method to retain the network structure information to a great extent is proved.

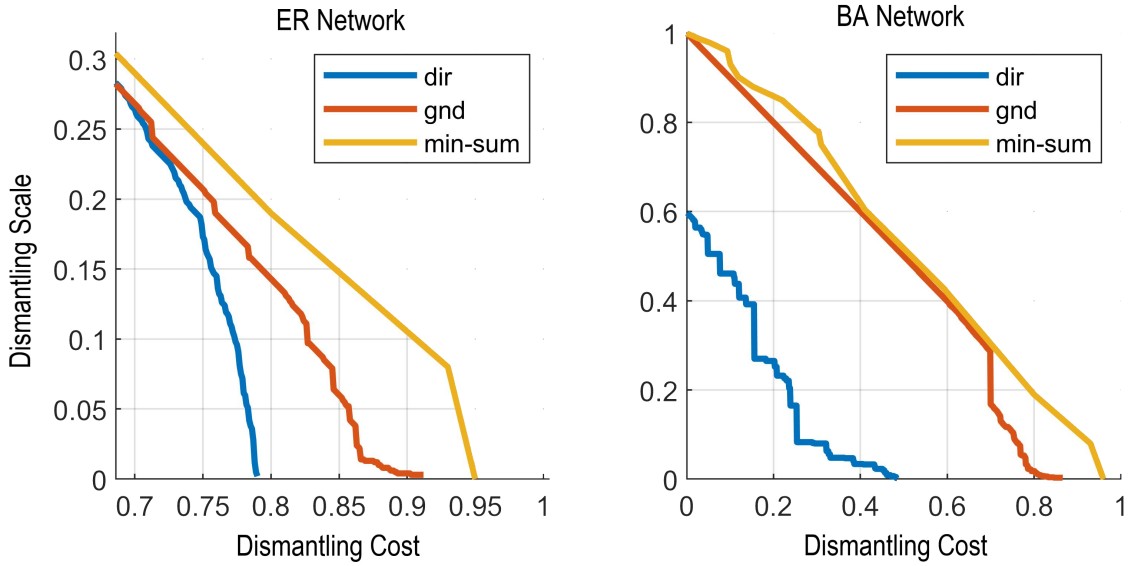

**Figure 2.** Curve graph of the disassembly cost and disassembly scale of directed ER random network and directed BA random network.

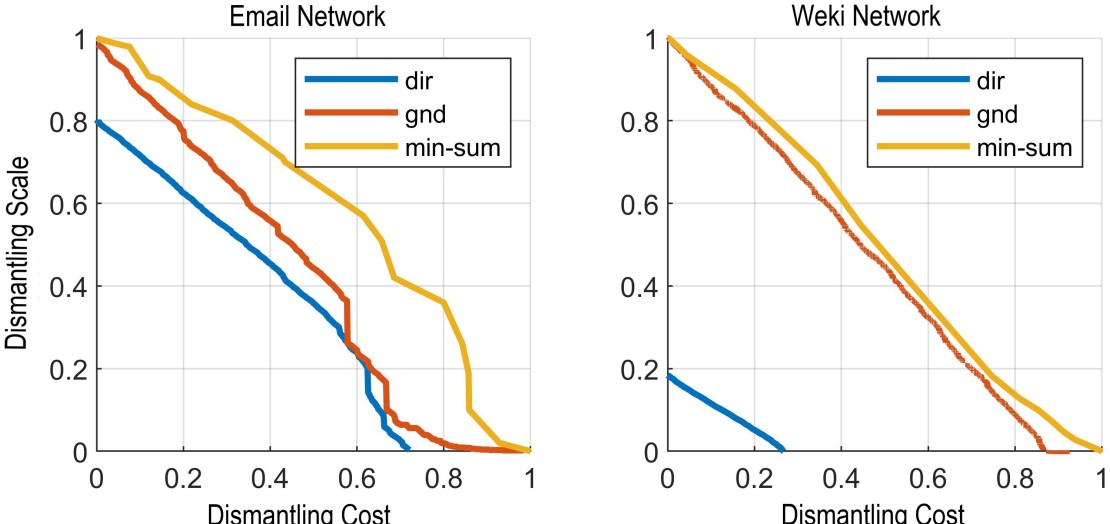

**Figure 3.** Curve graph of directed real network disassembly cost and disassembly scale.

The clustering coefficient in graph theory is used to measure the degree of node aggregation. There is evidence that in most real-world networks, especially in social networks, nodes tend to create relatively tightly connected groups; this possibility is often greater than the average probability of randomly establishing a relationship between two nodes. A network such as $G = (V, E)$ is formally composed of a set of nodes and edges between nodes, with edges connecting nodes. The neighborhood $N_i$ of a node $v_i$ is defined as its adjacent node, $N_i = \{v_j : e_{ij} \in E \vee e_{ji} \in E\}$. The local clustering coefficient $C_i$ of a node in a directed network is

$$C_i = \frac{\left|\left\{e_{jk} : v_j, v_k \in N_i, e_{jk} \in E\right\}\right|}{k_i(k_i - 1)} \quad (8)$$

As an alternative to the global clustering coefficient, Watt and Strogatz [19] use the average of local clustering coefficients of all vertices as the overall clustering level of the network.

$$C = \frac{1}{n}\sum_i C_i \quad (9)$$

Here, we compare the influence of the DIR and the other two disassembly methods on the degrees of node connection in the network by observing the change of the global clustering coefficient of the network during the disassembly process, and then explore the impact on the network structure.

The experiment first disassembles the artificial ER random network and the BA network; the relationship between the disassembly cost and the clustering coefficient is shown in Figure 4. When the disassembly cost is less than 0.7 in the artificial ER random network, the curve of the average clustering coefficient corresponding to the DIR method is more stable than the curve of the GND and Min-Sum method, and it also reaches a stable state first in the BA network. The DIR method has less disturbances for the clustering coefficient of the whole network compared to the GND and Min-Sum method, which reflects the superiority of removing overlapping nodes between modules by dividing the edge modules. The influence of the DIR method on the network structure is smaller than that of directly deleting nodes in the network; in the ER random network, the three methods will increase the network clustering coefficient with the disassembly in a certain period of time. This is because the disassembly has caused an increasing number of nodes in the network to appear in clusters. The result of the experiment in the real network is shown in Figure 5. It can be seen that in the real-world directed network, with the increase in the disassembly cost, the three disassembly methods will reduce the clustering coefficient in the network,

which is related to the sparsity of the real network. The dismantling of the nodes of the real network will reduce the agglomeration between nodes and the connection between groups will become sparse; however, it can still be seen from the graph that the curve corresponding to the DIR method is more gentle than the GND and Min-Sum methods. The influence of this aggregation phenomenon on the real network during the disassembly process is smaller than that of the two methods; and relatively speaking, the Min-Sum method will have a more obvious impact on the aggregation phenomenon of the network, because the Min-Sum method tends to remove nodes with large degrees in the network and is less able to protect the structural information of the network.

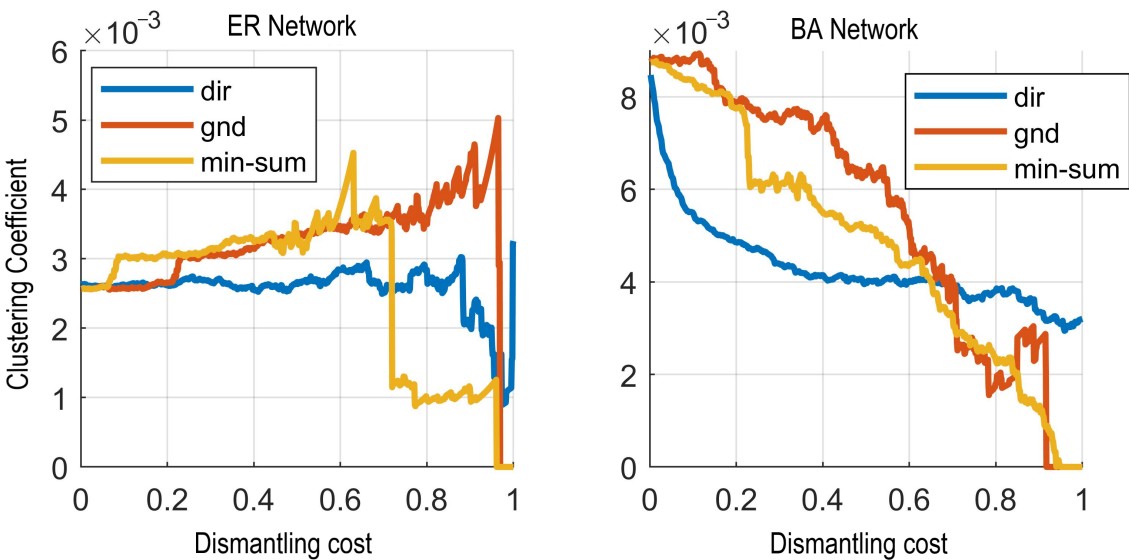

**Figure 4.** Curve graph of disassembly cost and clustering coefficient of directed ER random network and directed BA random network.

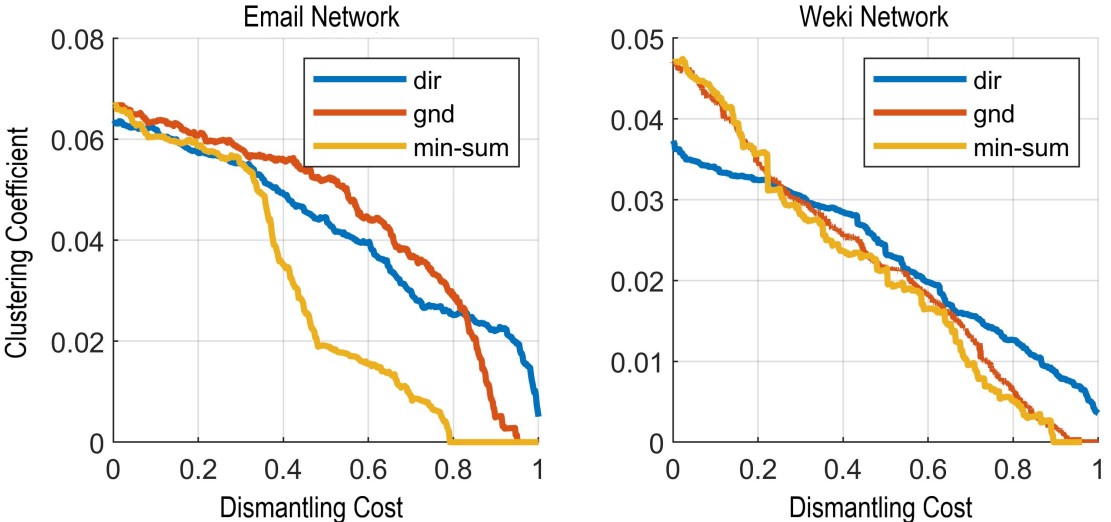

**Figure 5.** Curve graph of real network disassembly cost and clustering coefficient.

The coefficient of assortativity is used to measure whether the network is assortative or disassortative. It is used to investigate whether the nodes with similar values of degree in the network tend to be connected to its approximate nodes. It can be characterized by the Pearson coefficient $r$ (degree-degree correlation). $r > 0$ indicates that the entire network presents an assortative structure, and the nodes with large degrees tend to be connected to the nodes with large degrees. $r < 0$ indicates that the entire network presents

disassortativity, and *r* = 0 indicates that there is no correlation in the network structure. In the experiment, the change of the network structure by the dismantling of the DIR method is analyzed by observing the influence of the dismantling process on the network assimilation index.

As shown in Figures 6 and 7, the changes of the assortativity in the network of the DIR method are less in number than in the other two methods. Whether in the ER random network or in the real network, when the disassembly cost is less than 0.7, the blue curve is more gentle, the change of the global assortativity of the network is smaller, and the influence of removing the overlapping nodes between the edge modules on the assortativity of the network is smaller than that of the GND and Min-Sum methods.

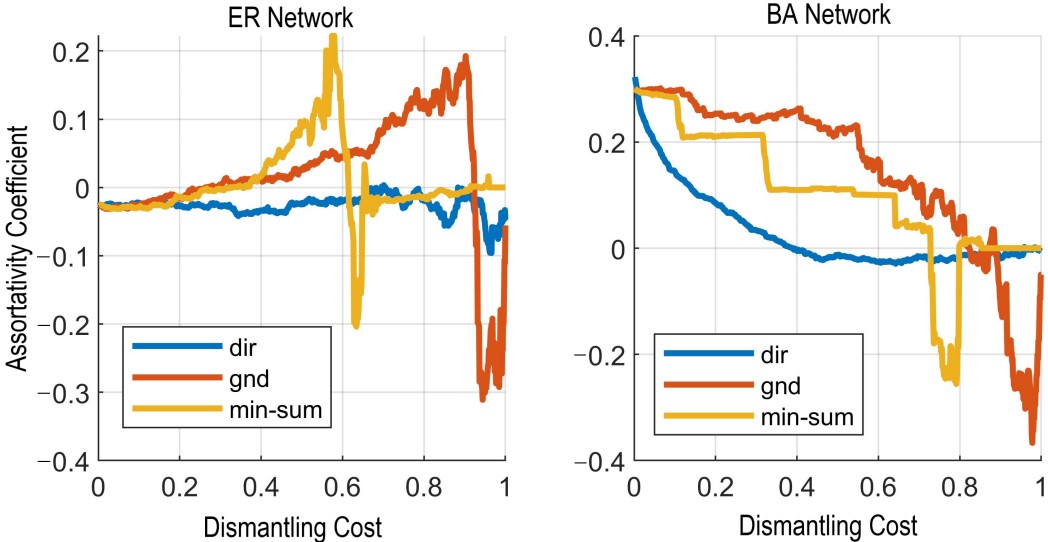

**Figure 6.** Curve graph of disassembly cost and assortative coefficient of directed ER random network and directed BA random network.

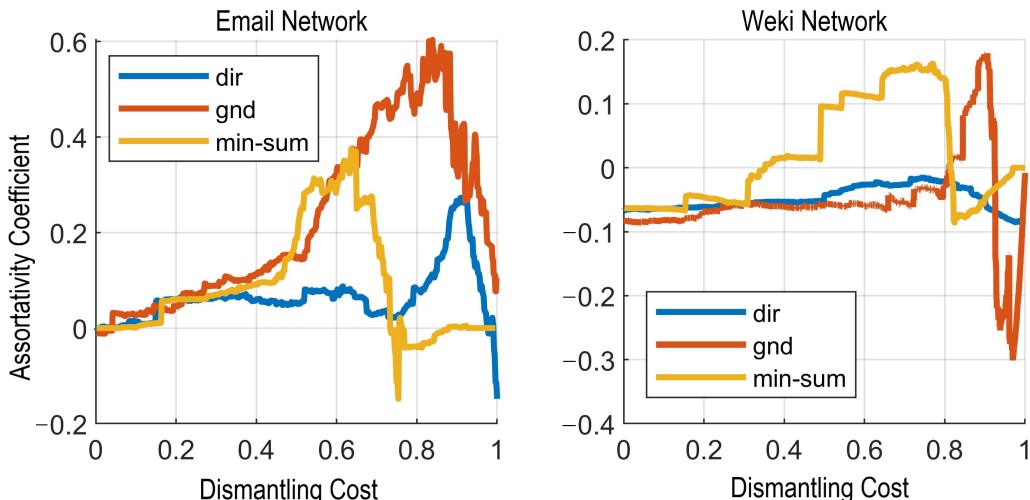

**Figure 7.** Curve graph of real network disassembly cost and assortativity coefficient.

The module degree [28] is used as a performance index to measure the community division. It is used to see the impact on the structure of the network community when we disassemble the network. The module degree function is $Q = \frac{1}{2} \sum_{i,j} a_{ij} \delta(c_i, c_j)$, where $a_{ij}$ is an element in the point adjacency matrix $A$, $c_i$, $c_j$ is the community to which node $i$ and $j$ belong to, and $\delta(c_i, c_j)$ is the membership function. If $i$ and $j$ are in the same community, it is 1, otherwise it is 0.

The calculation of the module degree $Q$ in the disassembly process of the DIR method is shown in Figure 8. It can be seen that when the disassembly cost is less than 0.8 in the picture, the module degree $Q$ increases with the increase in the disassembly cost. It shows that the removal node in the disassembly process also deletes the inter-group edges between different communities, which plays a certain role in promoting effective community division. When the cost is greater than 0.88, the disassembly will destroy the inter-group edges within the community and cause the modularity $Q$ to decrease sharply.

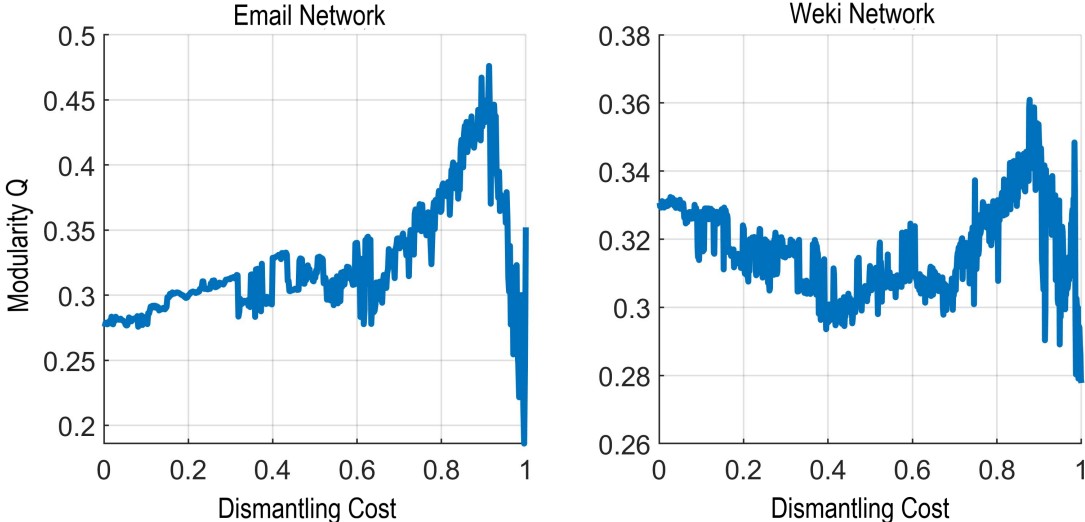

**Figure 8.** Curve graph of disassembly cost and module degree Q of real network.

In summary, the directed network disassembly (DIR) method we proposed in all the experiments has a higher disassembly efficiency than the GND and Min-Sum methods in both artificial directed networks and real directed networks. When the network is disassembled to the same scale, the DIR method incurs the lowest cost; at the same time, by comparing the clustering coefficient and the assortative coefficient in the disassembly process, it is also proved that the DIR method can reduce the influence of disassembly on the network clustering coefficient and the assortative coefficient in the disassembly process, and can also effectively retain the information in the network; the influence of the DIR method on network modularity is also explored through experiments. When the disassembly scale is less than a given threshold, the DIR method has a certain promoting effect on network community division.

## 6. Conclusions

An effective disassembly method is proposed for the disassembly of directed networks; the method combines edge module division with network disassembly, using a non-backtracking matrix to construct the function of the minimum number of edges for edge disassembly, and finds the overlapping nodes between edge modules to obtain the approximate solution of the eigenvector of the cost function. Different from the traditional undirected network disassembly method, the DIR method considers the unidirectional relationship between nodes in the directed network, makes full use of the excellent spectral characteristics of the non-backtracking matrix to divide the directed network, ensures the efficiency of disassembly and reduces the impact of disassembly on the overall structure of the network by removing the overlapping nodes between the edge modules during the network disassembly process. By comparing the DIR method with other methods in different artificial directed networks and real directed networks, it is proved that the DIR method is efficient in the network disassembly of directed networks. At the same time, it is also verified that the partition of the edge module applied to the application of non-backtracking operators in the network disassembly leads to a low disturbance of the network structure. The experimental results show that using this method to disassemble

the network can achieve lower costs and protect the structure information in the directed network to a great extent.

**Author Contributions:** Conceptualization, H.L. and J.M.; methodology, H.L.; software, H.L.; validation, J.M. and P.W.; supervision, H.L. and J.M.; formal analysis, J.M. and P.W.; writing—original draft preparation, H.L. and P.W.; funding acquisition, H.L. and J.M. All authors have read and agreed to the published version of the manuscript.

**Funding:** This research was funded by National Natural Science Foundation of China under Grant 71871233, Science and Technology Project of Hebei Education Department under Grant ZD2022031, Research on the Development of Social Science of Hebei Province under Grant 20220202181 and Fundamental Research Funds for the Hebei Universities under Grant 2021YWF08.

**Institutional Review Board Statement:** Not appliable.

**Informed Consent Statement:** Not appliable.

**Data Availability Statement:** Not appliable.

**Conflicts of Interest:** The authors declare no conflict of interest. The funders had no role in the design of the study; in the collection, analyses, or interpretation of data; in the writing of the manuscript; or in the decision to publish the results.

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
