# Peer review of "Directed Network Disassembly Method Based on Non-Backtracking Matrix"

_applsci, doi:10.3390/app122312047_

Round 1

Reviewer 1 Report

The article addresses an interesting research topic. However, the quality of presentation and the English language prevent, in my opinion, a detailed evaluation of the results.

The authors should clarify the statement on Page 1, Lines 24-26. Please, make the relation between the method with the cited works and problems. In my opinion, it is currently confusing. For instance, how does removing/disabling connect with the previous argument?

The authors state: "Experiments have proved 29 that this method can effectively immune the spread of epidemics in the population". What method?

The authors could rewrite the introduction by following a more clear sequence of argumentations. For instance, contextualization, explanation of the general method, examples of applications (are references [7], [8], and [9] such examples?), problem description, proposed solution, and organization of the paper.

The second paragraph of the introduction should be split up. It is a too-long paragraph, decreasing readability.

The example of virus transmission at the end of Page 1 is interesting. However, the clarity of the explanation (e.g., English) should be improved. Besides, is it an example from a previously published study? If so, the citation is missing.

Is paragraph 2 also presenting related works (and limitations)? To improve readability and clarity, a related works section should be included after the introduction (a new Section 2).

The authors state, "Most of the existing disassembly algorithms are carried out in undirected networks, 75 while there are few disassembly algorithms for directed networks.". The authors should discuss these studies (along with limitations) in the new related works section.

The last paragraph of the introduction is also too long, decreasing readability and clarity.

Please, define the DIR acronym at the first mention. Please, also review the remaining acronym to ensure the definition is in the correct order.

Section 2 presents underlying methods and the proposed solution. However, in my opinion, the authors should provide a previous section with background information to support readers in understanding the methods underlying the proposed solution. This can also improve readability and clarity of presentation. Besides, the authors could change the title of Section 2 (e.g., Proposed Solution or Proposed Method). Therefore, the article could be structured as follows. Section 1 (introduction), Section 2 (related works), Section 3 (preliminaries), Section 4 (proposed method), Section 5 (experimental results), Section 6 (discussion), and Section 7 (conclusion).

The explanation regarding Figure 1 is not completely clear.

The authors can also better justify the chosen datasets (e.g., regarding size) and the possible impacts on conclusions.

Some examples of simple issues:

proposed, this method --> proposed. This method

random networks, this --> random networks. This

( such as --> ( e.g.,

The data set --> The dataset

The English need revision, mainly regarding punctuation.

Author Response

General Comment: The article addresses an interesting research topic. However, the quality of presentation and the English language prevent, in my opinion, a detailed evaluation of the results.

[Response] Thank you for all of your detailed comments and suggestions. We found them very constructive as we approached our revision. In the following sections, you will find our responses to each of your points and suggestions. We are grateful for the time and energy you expended on our behalf.

[Comment 1] The authors should clarify the statement on Page 1, Lines 24-26. Please, make the relation between the method with the cited works and problems. In my opinion, it is currently confusing. For instance, how does removing/disabling connect with the previous argument?

[Response] Thank you for your careful review and valuable suggestions. According to your opinion, we have revised the manuscript ,making the relation between the method with the cited works and problems.(Introduction section, lines 26-29)

[Comment 2] The authors state: "Experiments have proved 29 that this method can effectively immune the spread of epidemics in the population". What method?

[Response] We are very sorry for our negligence and have revised the sentence. Accordingly, we have modified the sentence in the revised version.(Introduction section, lines 30-31)

[Comment 3] The authors could rewrite the introduction by following a more clear sequence of argumentations. For instance, contextualization, explanation of the general method, examples of applications (are references [7], [8], and [9] such examples?), problem description, proposed solution, and organization of the paper.

[Response] Thanks for your valuable comments and helpful references. Based on your suggestion, we have rewritten the introduction by following a clearer sequence of argument.

[Comment 4] The second paragraph of the introduction should be split up. It is a too-long paragraph, decreasing readability.

[Response] Thanks for raising this important point. Accordingly, we have split the second paragraph into two shorter paragraphs to increase readability.

[Comment 5] The example of virus transmission at the end of Page 1 is interesting. However, the clarity of the explanation (e.g., English) should be improved. Besides, is it an example from a previously published study? If so, the citation is missing.

[Response] We are very sorry for missing the ciation. This is indeed an example of a previously published study, and we have added this ciation in the revised version.(Introduction section, line 38)

[Comment 6] Is paragraph 2 also presenting related works (and limitations)? To improve readability and clarity, a related works section should be included after the introduction (a new Section 2).

[Response] Thanks for your valuable comments and helpful references. Paragraph 2 does present the relevant work, and we have added a new section after the introduction to introduce the related works to improve readability and clarity.

[Comment 7] The authors state, "Most of the existing disassembly algorithms are carried out in undirected networks, 75 while there are few disassembly algorithms for directed networks.". The authors should discuss these studies (along with limitations) in the new related works section.

[Response] Thanks for your constructive and valuable comments. We have discussed these studies along with limitations in the new related works section.

[Comment 8] The last paragraph of the introduction is also too long, decreasing readability and clarity.

[Response] Thank you for raising this important point. As you said, the last paragraph of the introduction is too long, so we have split the second paragraph into two shorter paragraphs to increase readability and clariey.

[Comment 9] Please, define the DIR acronym at the first mention. Please, also review the remaining acronym to ensure the definition is in the correct order.

[Response] Thank you for your detailed suggestions. We are very sorry for our negligence, and we have defined the DIR acronym at the first mention as well as the remaining acronym to ensure the definition is in the correct order.(Introduction section, lines 57-58, line 60, line 85)

[Comment 10] Section 2 presents underlying methods and the proposed solution. However, in my opinion, the authors should provide a previous section with background information to support readers in understanding the methods underlying the proposed solution. This can also improve readability and clarity of presentation. Besides, the authors could change the title of Section 2 (e.g., Proposed Solution or Proposed Method). Therefore, the article could be structured as follows. Section 1 (introduction), Section 2 (related works), Section 3 (preliminaries), Section 4 (proposed method), Section 5 (experimental results), Section 6 (discussion), and Section 7 (conclusion).

[Response] Thank you for your valuable suggestions. We have provided a previous section with background information to support readers in understanding the methods underlying the proposed solution and to improve readability and clarity of presentation. Besides, we have restructured the article in the order you suggest.

[Comment 11] The explanation regarding Figure 1 is not completely clear.

[Response] Thank you for carefully and patiently reviewing of our manuscript. We agree with your comment, and we are very sorry for our negligence and have modified the sentences in the revised version (Preliminaries section, paragraph 2).

[Comment 12] The authors can also better justify the chosen datasets (e.g., regarding size) and the possible impacts on conclusions.

[Response] Thanks for your constructive and valuable comments. The reason for choosing artificial networks with 1000 nodes and an average degree of 10 is that such networks are closer to real networks, such as the email network between members of Rovira i Virgili University and the UC Irvine student forum network. Selecting such a network for experiments can make the experimental results more general.

Again, we appreciate all of your insightful comments. We worked hard to be responsive to them. Thank you for taking the time and energy to help us improve the paper.

Reviewer 2 Report

Comments of the manuscript ID: applsci-1938501 :

Title: “Directed Network Disassembly Method Based on Non - backtracking Matrix ” 

Comments:

\1. The authors propose a new method for dismantling networks, oriented explicitly to directed networks in which other traditional methods assume that there is no directionality in the edges. The methodology uses the non-backtracking matrix as input to initiate the dismantling process. 

\2. As indicated by the authors, the few existing methods for dismantling networks are focused on directed networks and even less on directed networks, so the contribution of this work focuses precisely on a solution to find subsets of disconnected graphs in which the original degree contains edges with direction. I assume that for that reason, the comparison of the proposed method is performed only with two other methods (gnd and min-sum methods).

\3. The choice of networks to compare the proposed method with others seems appropriate (ER random net, BA random net, and two other real nets). However, in these four cases, the sizes of the networks are pretty similar (about 1000 nodes and one of about 8000 nodes). I wonder, Is the proposed method able to deal with larger networks? Is the method able to deal with networks coming from different domains?; for example, Ren at al., (2019) apply the gnd method on networks of millions of nodes. I raise this question, not necessarily for the authors to test with an large number of other networks, but to consider the generability of the results on networks of larger sizes and from very different domains. A similar concern is raised by Wandelt et al., (2018).

\4. It is unclear what definition the authors take for the concept of dismantling cost. I have assumed it is the same as the one used in Ren et al., (2019). Perhaps it would be helpful to clarify this definition in the manuscript.

\5. Regarding the analysis of the impact of different dismantling methods on network structure: In Figures 2 through 7, we observe the evolution of specific network characteristics (clustering coef. assortativity, etc.,,,) under different dismantling costs. If I understand correctly, at the beginning when dismantling costs are zero, we have the original network; consequently, the network characteristic corresponds to the original value. The idea is that as we dismantle the network (higher cost), the network characteristic remains (ideally) unchanged. This means a low impact of dismantling. It is inevitable that the process will destroy the original value of the clustering coefficient or assortativity at some point. So, under this logic, I observe that the proposed method generates a higher impact on assortativity and cluster coef. concerning the other two methods in BA networks, but not for the other type of networks. Is there any reason for this?

Author Response

[Comment 1] The authors propose a new method for dismantling networks, oriented explicitly to directed networks in which other traditional methods assume that there is no directionality in the edges. The methodology uses the non-backtracking matrix as input to initiate the dismantling process.

[Response] Thank you for your careful review and valuable suggestions. Just as you said, we propose a new method of network disassembly for directed networks, and the methodology uses the non-backtracking matrix as input to initiate the dismantling process.

[Comment 2] As indicated by the authors, the few existing methods for dismantling networks are focused on directed networks and even less on directed networks, so the contribution of this work focuses precisely on a solution to find subsets of disconnected graphs in which the original degree contains edges with direction. I assume that for that reason, the comparison of the proposed method is performed only with two other methods (gnd and min-sum methods).

[Response] Thanks for raising this important point. As you pointed out, most of the current methods of dismantling networks focus on undirected networks, and little attention is paid to directed networks. For this reason, the comparison of the proposed method is performed only with the GND method and the Min-Sum method.

[Comment 3] The choice of networks to compare the proposed method with others seems appropriate (ER random net, BA random net, and two other real nets). However, in these four cases, the sizes of the networks are pretty similar (about 1000 nodes and one of about 8000 nodes). I wonder, Is the proposed method able to deal with larger networks? Is the method able to deal with networks coming from different domains?; for example, Ren at al., (2019) apply the gnd method on networks of millions of nodes. I raise this question, not necessarily for the authors to test with an large number of other networks, but to consider the generability of the results on networks of larger sizes and from very different domains. A similar concern is raised by Wandelt et al., (2018).

[Response] You raise an important point with respect to data processing capabilities. The proposed method is able to deal with larger networks and networks coming from different domains. The time complexity of our method is lower than other algorithms, so the data set that can be processed is larger. In future work, we will focus on dealing with larger networks.

[Comment 4] It is unclear what definition the authors take for the concept of dismantling cost. I have assumed it is the same as the one used in Ren et al., (2019). Perhaps it would be helpful to clarify this definition in the manuscript.

[Response] We are very sorry for our negligence and have carefully reviewed the relevant parts. We have clarified the definition of disassembly cost in the revised version.(Introduction section, paragraph 3, lines 47-49, Experimental Results section, paragraph 2, lines 266-268 ) In this paper, we consider the case where the disassembly cost is the unit cost ( i.e. the number of nodes ). This is different from the concept mentioned in Ren et al. ' s article. The method proposed by Ren et al.considers that the disassembly cost of the node is equal to the node weight and is a non-unit cost.

[Comment 5] Regarding the analysis of the impact of different dismantling methods on network structure: In Figures 2 through 7, we observe the evolution of specific network characteristics (clustering coef. assortativity, etc.,,,) under different dismantling costs. If I understand correctly, at the beginning when dismantling costs are zero, we have the original network; consequently, the network characteristic corresponds to the original value. The idea is that as we dismantle the network (higher cost), the network characteristic remains (ideally) unchanged. This means a low impact of dismantling. It is inevitable that the process will destroy the original value of the clustering coefficient or assortativity at some point. So, under this logic, I observe that the proposed method generates a higher impact on assortativity and cluster coef. concerning the other two methods in BA networks, but not for the other type of networks. Is there any reason for this?

[Response] Thank you very much for proposing a point that is easily overlooked. An artificial directed BA network contains a small number of hub nodes. Compared with the other two methods, our proposed method is more likely to delete these hub nodes so that the degree of clustering and assortativity of the network decreases rapidly. Therefore, the proposed method has a higher impact on the assortativity and clustering coefficients of the other two methods in the BA network.

We did our best to improve the manuscript and made some modifications. We thank the reviewers for their enthusiastic work and hope that the correction will be approved. Again, thank you very much for your comments and suggestions.